# Bayesian Active Learning with Fully Bayesian Gaussian Processes

**Christoffer Riis**     **Francisco Antunes**     **Frederik Boe Hüttel**
**Carlos Lima Azevedo**     **Francisco Câmara Pereira**
DTU Management
Machine Learning for Smart Mobility
`{chrrii,franant,fbohy,climaz,camara}@dtu.dk`

## Abstract

The bias-variance trade-off is a well-known problem in machine learning that only gets more pronounced the less available data there is. In active learning, where labeled data is scarce or difficult to obtain, neglecting this trade-off can cause inefficient and non-optimal querying, leading to unnecessary data labeling. In this paper, we focus on active learning with Gaussian Processes (GPs). For the GP, the bias-variance trade-off is made by optimization of the two hyperparameters: the length scale and noise-term. Considering that the optimal mode of the joint posterior of the hyperparameters is equivalent to the optimal bias-variance trade-off, we approximate this joint posterior and utilize it to design two new acquisition functions. The first is a Bayesian variant of Query-by-Committee (B-QBC), and the second is an extension that explicitly minimizes the predictive variance through a Query by Mixture of Gaussian Processes (QB-MGP) formulation. Across six simulators, we empirically show that B-QBC, on average, achieves the best marginal likelihood, whereas QB-MGP achieves the best predictive performance. We show that incorporating the bias-variance trade-off in the acquisition functions mitigates unnecessary and expensive data labeling.

## 1 Introduction

Gaussian Processes (GPs) are well-known for their ability to deal with small to medium-size data sets by balancing model complexity and regularization [Williams and Rasmussen, 2006]. Together with their inherent ability to model uncertainties, this has made GPs the go-to models to use for Bayesian optimization and metamodeling [Snoek et al., 2012, Gramacy, 2020]. For both cases, the data is often scarce, making the modeling task a balance between complexity and regularization, i.e., preventing severe overfitting while maintaining the ability to fit nonlinear functions. Likewise, the ability to quantify the uncertainty often guides the acquisition functions of Bayesian optimization and active learning schemes, which are inevitably required to build metamodels efficiently.

On the other hand, it is not flawless to use GPs in Bayesian optimization and active learning. In both cases, the same GP is used in an iterative process firstly to predict the mean and variance of a new data point and then secondly to use those estimates to guide the data acquisition. Since this is an iterative process, poor predictions will lead to poor data acquisition, and vice versa. The problem is less pronounced for larger data sets as predictions become increasingly accurate as more data is available. However, in Bayesian optimization and active learning, where the data sets tend to be relatively small, wrong predictions can result in misguidance, thus hindering performance and efficiency. In this paper, we mitigate this problem by applying a fully Bayesian approach to the GPs and formulating two new acquisition functions for active learning. Where a single GP trained with a maximum likelihood estimate only represents one model hypothesis, a Fully Bayesian Gaussian Process (FBGP) represents multiple model hypotheses at once. We will utilize this extra information

36th Conference on Neural Information Processing Systems (NeurIPS 2022).

to create an acquisition function that simultaneously seeks the best model hypothesis and minimize the prediction error of the GP.

The hyperparameters of a GP are typically fitted through evaluation of the marginal likelihood, which automatically incorporates a trade-off between complexity and regularization [Williams and Rasmussen, 2006], also known as the bias-variance trade-off [Bishop, 2006]. However, when the data is scarce, it is more challenging to choose the appropriate trade-off, and different configurations of the hyperparameters of the GP can give rise to distinct fits. We highlight this issue in Figure 1, where two seemingly reasonable fits describe the data very distinctly, which would result in different acquisitions of new data.

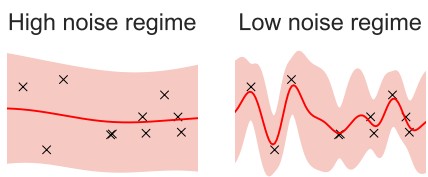

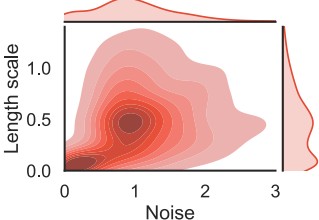

Figure 1: Two GPs with different hyperparameters. The left plot shows a GP with high noise and a long length scale, and the right plot shows a GP with low noise and a short length scale.

Figure 2: Joint posterior of the two hyperparameters length scale and noise. The posterior is multimodal with low noise, high signal, and high noise, low signal mode.

**The bias-variance trade-off for Gaussian Processes**   For a GP with a common stationary co-variance function with a length scale and a noise-term, e.g., the radial-basis function or the Matérn class of functions with a Gaussian likelihood, the challenge of choosing the best trade-off between bias-variance trade-off, can be formulated using the hyperparameters of the GP. The two central hyperparameters are the length scale $\ell$ and the variance of the noise $\sigma_\varepsilon^2$. The length scale describes how much the underlying model $f$ fluctuates, i.e., if the length scale is short or long, the model varies quickly or slowly, respectively. In terms of tuning the hyperparameters, often a short length scale goes well with a small noise-term since the greater flexibility of a short length scale means that the noise level can be reduced. This results in a flexible model with high variance and low bias. Conversely, a long length scale tends to increase the noise level, resulting in a rigid model with low variance and high bias [Bishop, 2006]. In the extremes, the former corresponds to a white-noise process model, and the latter corresponds to a constant with white noise [Williams and Rasmussen, 2006]. Thus, there is a direct relation between the bias-variance trade-off and the values of the two hyperparameters.

**How to choose the best bias-variance trade-off?**   Finding a good bias-variance trade-off is a non-trivial problem. In the case of small to medium-sized data sets, the joint posterior distribution of the two hyperparameters is likely to be characterized by two modes, as illustrated in Figure 2 for the data in Figure 1. The two modes illustrate two different bias-variance trade-offs, which both describe the data well. However, depending on the choice of trade-off, the data acquisition can be very distinct, and thus a wrong choice of mode will imply non-optimal guidance from the acquisition function.

Though the multimodal posterior has been studied for GPs before [Yao et al., 2022], the literature often searches for the single best mode with clever initializations of the hyperparameters [Williams and Rasmussen, 1995] or by favoring small $\ell$ and $\sigma_\varepsilon^2$ by either always initializing hyperparameters in the low noise regime or by applying strong priors [Gramacy, 2020]. However, none of these approaches directly address the core problem: *which mode to choose*? When working with Bayesian optimization and active learning, this should ideally be answered with prior information about the problem, although typically that is not available, making these approaches less practical [Antunes et al., 2018, Gramacy, 2020, Riis et al., 2021].

**Our contribution**   We follow a general approach to the problem and assume no prior knowledge about the functional form of the data, kernel, nor hyperparameters. If it is known that the data has a periodic trend or high noise, it is advantageous to incorporate that into the kernel and the hyperparameters by using a periodic kernel and a prior on the hyperparameters that favor high noise, respectively. However, with no prior knowledge, we tend to use the general-purpose Radial-

Basis function (RBF) kernel with non-informative priors on the hyperparameters. Further, the fitted hyperparameters are often given by point estimates (e.g. fitted with maximum likelihood estimation or maximum a posteriori), but we consider multiple model hypotheses by replacing the fitting procedure of the marginal likelihood with Markov Chain Monte Carlo (MCMC) sampling to get the joint posterior of the hyperparameters. The result is that we have multiple models (same kernel, but different hyperparameters), which represent different model hypotheses. We utilize the extra information from the hyperparameters' joint posterior to handle the bias-variance trade-off by incorporating the extra information into two new acquisition functions.

Our main contribution is the proposal of two new acquisition functions for active learning that utilize the extra information from the hyperparamters' posterior estimated by MCMC to seek the most reasonable mode alongside minimizing the predictive variance. Through empirical results, we show that the two acquisition functions are more accurate and robust than other common functions across multiple benchmark simulators used in the literature.

## 2 Related work

In this section, we review related work to the proposed acquisition functions. We cover active learning schemes for regression tasks, including Query-by-Committee and GP as Gaussian Mixture Models.

**Active Learning** The main idea of active learning is to *actively* choose a new data point to label and add to the current training data set, to iteratively improve the performance of the predictive model [Settles, 2009]. In the context of metamodeling or surrogate modeling of simulators, new data is often added sequentially, i.e., one data point at a time [Gramacy, 2020], but in other applications, it can be beneficial to query batches of data instead [Kirsch et al., 2019].

The acquisition functions can be divided into model-based and model-free functions, where the former utilize information from the model and is often based on uncertainty measures (recently also function values and gradients [Fernandez et al., 2020, Svendsen et al., 2020]), whereas the latter do not use information from the model and is typically based on distance metrics in the input space [O'Neill et al., 2017]. Both types of functions seek to minimize the expected predictive loss of the model. Another distinction between the acquisition functions is decision-based and information theory-based [Houlsby et al., 2011]. Decision-based functions seek to minimize the expected predictive loss in the hope of maximizing the performance on the test set. Information-theoretic-based functions instead try to reduce the number of possible models, e.g., through the KL-divergence or Shannon entropy.

It is not straightforward to use information-theoretic acquisition functions. However, if one has access to the posterior of the parameters, Houlsby et al. [2011] have derived the algorithm *Bayesian Active Learning by Disagreement* (BALD), which can be applied in general. Generally, BALD seeks the data point that maximizes the decrease in the expected posterior entropy of the parameters.

**Query-by-Committee** The Query-by-Committee (QBC) is a specific acquisition function that was originally proposed for classification tasks [Seung et al., 1992]. It aims to maximize the disagreement among the committee to get the highest information gain and minimize the version space, which is the set of model hypotheses aligned with the training data. The construction of the committee is the core component of QBC since it is the committee's ability to accurately and diversely represent the version space that gives rise to informative disagreement criteria [Settles, 2009].

Query-by-Committee can also be applied for regression problems. Krogh and Vedelsby [1995] construct the members of the committee by random initializations of the weights in the neural networks. RayChaudhuri and Hamey [1995] apply bagging and train the members on different subsets of the data set. In general, QBC constructed by bagging has been used as a benchmark with mixed results [Cai et al., 2013, Wu, 2018, Wu et al., 2019]. Burbidge et al. [2007] show that the less noise there is in the output, the better QBC is compared to random querying. They also highlight the fact that with a misspecified model, QBC might perform worse than random querying. None of these approaches explore the usage of MCMC samples of the posterior to construct a committee.

**Gaussian Process as a Gaussian Mixture Model** Mixture models have recently been applied in active learning for classification tasks. Iswanto [2021] proposes to use Gaussian Mixture Models (GMMs) with active learning, where he designs a specific acquisition function that queries the data point that maximizes the expected likelihood of the model. Zhao et al. [2020] use a mixture of GPs in active learning, where each component is fitted to a subset of the training set. The combination of

GMMs and GPs have previously been explored for static data sets. Chen and Ren [2009] investigate regression tasks and apply bagging, where they repeatedly randomly sample data points from the training set to construct new subsets to get GPs fitted to different data.

## 3 Gaussian Processes

The Gaussian Processes (GPs) are the central models in this work. In this section, we give a brief overview of GPs before covering the Fully Bayesian GPs. For a thorough description of GPs, we refer to Williams and Rasmussen [2006].

A Gaussian Process (GP) is a stochastic function fully defined by a mean function $m(\cdot)$ and a covariance function (often called a kernel) $k(\cdot, \cdot)$. Given the data $\mathcal{D} = (X, \boldsymbol{y}) = \{\boldsymbol{x}_i, y_i\}_{i=1}^N$, where $y_i$ is the corrupted observations of some latent function values $\boldsymbol{f}$ with Gaussian noise $\varepsilon$, i.e., $y_i = f_i + \varepsilon_i$, $\varepsilon_i \in \mathcal{N}(0, \sigma_\varepsilon^2)$, a GP is typically denoted as $\mathcal{GP}(m_{\boldsymbol{f}}(\boldsymbol{x}), k_{\boldsymbol{f}}(\boldsymbol{x}, \boldsymbol{x}'))$. It is common practice to set the mean function equal to the zero-value vector and thus, the GP is fully determined by the kernel $k_{\boldsymbol{f}}(\boldsymbol{x}, \boldsymbol{x}')$. For short, we will denote the kernel $K_{\boldsymbol{\theta}}$, which explicitly states that the kernel is parameterized with some hyperparameters $\boldsymbol{\theta}$. The generative model of the GP can be found in Appendix A.1. Given the hyperparameters $\boldsymbol{\theta}$, the predictive posterior for unknown test inputs $X^\star$ is given by $p(\boldsymbol{f}^\star | \boldsymbol{\theta}, \boldsymbol{y}, X, X^\star) = \mathcal{N}(\boldsymbol{\mu}^\star, \boldsymbol{\Sigma}^\star)$ with

$$\boldsymbol{\mu}^\star = K_{\boldsymbol{\theta}}^\star \left(K_{\boldsymbol{\theta}} + \sigma_\varepsilon^2 \mathbb{I}\right)^{-1} y \quad \text{and} \quad \boldsymbol{\Sigma}^\star = K_{\boldsymbol{\theta}}^{\star\star} - K_{\boldsymbol{\theta}}^\star \left(K_{\boldsymbol{\theta}} + \sigma_\varepsilon^2 \mathbb{I}\right)^{-1} K_{\boldsymbol{\theta}}^{\star\top} \tag{1}$$

where $K_{\boldsymbol{\theta}}^{\star\star}$ denotes the covariance matrix between the test inputs, and $K_{\boldsymbol{\theta}}^\star$ denotes the covariance matrix between the test inputs and training inputs.

We use the canonical kernel automatic relevance determination (ARD) Radial-basis function (RBF) given by $k(\boldsymbol{x}, \boldsymbol{x}') = \exp\left(-||\boldsymbol{x} - \boldsymbol{x}'||^2 / 2\boldsymbol{\ell}^2\right)$ where $\boldsymbol{\ell}$ is a vector of length scales $\ell_1, ..., \ell_d$, one for each input dimension. Often the kernel is scaled by an output variance but here we fix it to one and solely focus on the two other hyperparameters: length scale and noise-term. The noise-term $\sigma_\varepsilon^2$ is integrated into the kernel with an indicator variable by adding the term $\sigma_\varepsilon^2 \mathbb{I}_{\{\boldsymbol{x}=\boldsymbol{x}'\}}$ to the current kernel [Williams and Rasmussen, 2006, Bishop, 2006].

**Fully Bayesian Gaussian Processes (FBGP)** An FBGP extends a GP by putting a prior over the hyperparameters $p(\boldsymbol{\theta})$ and approximating their full posteriors. The joint posterior is then given by

$$p(\boldsymbol{f}, \boldsymbol{\theta} | \boldsymbol{y}, X) \propto p(\boldsymbol{y} | \boldsymbol{f}) p(\boldsymbol{f} | \boldsymbol{\theta}, X) p(\boldsymbol{\theta}) \tag{2}$$

and the predictive posterior for the test inputs $X^\star$ is

$$p(\boldsymbol{y}^\star | \boldsymbol{y}) = \iint p\left(\boldsymbol{y}^\star | \boldsymbol{f}^\star, \boldsymbol{\theta}\right) p(\boldsymbol{f}^\star | \boldsymbol{\theta}, \boldsymbol{y}) p(\boldsymbol{\theta} | \boldsymbol{y}) d\boldsymbol{f}^\star d\boldsymbol{\theta} \tag{3}$$

where the conditioning on $X$ and $X^\star$ have been omitted for brevity. The inner integral reduces to the predictive posterior given by a normal GP, whereas the outer integral remains intractable and is approximated with MCMC inference with $M$ samples as

$$p\left(\boldsymbol{y}^\star | \boldsymbol{y}\right) = \int p\left(\boldsymbol{y}^\star | \boldsymbol{y}, \boldsymbol{\theta}\right) p(\boldsymbol{\theta} | \boldsymbol{y}) d\boldsymbol{\theta} \quad \approx \quad \frac{1}{M} \sum_{j=1}^M p\left(\boldsymbol{y}^\star | \boldsymbol{y}, \boldsymbol{\theta}_j\right), \quad \boldsymbol{\theta}_j \sim p(\boldsymbol{\theta} | \boldsymbol{y}) \tag{4}$$

Adapting the hyperparameters of an FBGP is computationally expensive compared to the approach with GPs and maximum likelihood estimation. However, in Bayesian optimization and active learning, the computational burden for querying a new data point will often be of magnitudes higher. For example for simulators, the computational cost of querying a new data point is, in general, expensive and can take minutes and hours [Gorissen et al., 2009, Riis et al., 2021, Chabanet et al., 2021].

## 4 Active Learning

In this section, we lay out the most common acquisition functions and then propose first a Bayesian variant of Query-by-Committee and second an extension motivated by Gaussian Mixture Models, which seek to minimize both the predictive variance and the number of model hypotheses.

Many active learning acquisition functions are based on the model's uncertainty and entropy and can thus be denoted as *Bayesian* active learning acquisition functions [Settles, 2009, Gramacy, 2020]. The most common acquisition function is based on the predictive entropy and denoted *Active Learning MacKay* (ALM) [MacKay, 1992]. All the following objective functions query a new data point by maximizing the argument $x$. In the following, we write a new test point $x^\star$ as $x$ for brevity. All the acquisition functions choose a data point $x$ among the possible data points in the unlabeled pool $U$.

**Entropy (ALM)**   For a Gaussian distribution, the Shannon entropy $H[\cdot]$ is proportional to the predictive variance $\sigma^2(x)$ (derived in A.2), so ALM is given as

$$\text{ALM} = H[y|x, \mathcal{D}] \propto \sigma^2(x) \tag{5}$$

Intuitively, ALM queries the data point, where the uncertainty of the prediction is the highest.

If we have access to the posterior of the model's hyperparameters, we can utilize acquisition functions with an extra Bayesian level. The posterior of the hyperparameters of an FBGP can be estimated with MCMC such that GPs with different kernel parameters can be drawn, e.g., the length scale and the noise-term $\ell, \sigma_\varepsilon^2 \sim p(\theta|\mathcal{D})$. The following four acquisition functions all utilize this information and are approximated using the samples from the MCMC, cf. (4).

**Entropy (B-ALM)**   With the information from the posterior $p(\theta|\mathcal{D})$, the criteria for the extra Bayesian variant of ALM (B-ALM) is then given as

$$\text{B-ALM} = H\left[\int p(y|x, \theta)p(\theta|\mathcal{D})d\theta\right] = H\left[\mathbb{E}_{p(\theta|\mathcal{D})}[p(y|x, \theta)]\right] \propto \mathbb{E}_{p(\theta|\mathcal{D})}[\sigma_\theta^2(x)|\theta] \tag{6}$$

**Bayesian Active learning by Disagreement (BALD)**   Another common objective in Bayesian active learning, is to maximize the expected decrease in posterior entropy [Guestrin et al., 2005, Houlsby et al., 2012]. Houlsby et al. [2011] rewrite the objective from computing entropies in the parameter space to the output space by observing that it is equivalent to maximizing the conditional mutual information between the model's parameters $\hat{\theta}$ and output $I[\hat{\theta}, y|x, \mathcal{D}]$. The acquisition function is denoted Bayesian Active Learning by Disagreement (BALD) and the criteria is given by:

$$I[\hat{\theta}, y|x, \mathcal{D}] = H[y|x, \mathcal{D}] - \mathbb{E}_{p(\hat{\theta}|\mathcal{D})}[H[y|x, \hat{\theta}]] \tag{7}$$

BALD was originally derived for non-parametric discriminative models but has recently been extended to batches and deep learning with great success [Kirsch et al., 2019]. In the context of non-parametric models, such as GPs, the models parameters now correspond to the latent function $f$. For a regular GP, BALD is equivalent to ALM (cf. A.3), although that is not the case for an FBGP. If we let the hyperparameters be the main parameters of interest and set $f$ to be a nuisance parameter, BALD can be written as (cf. A.3.1):

$$\text{BALD} = H\left[\mathbb{E}_{p(\theta|\mathcal{D})}[y|x, \mathcal{D}, \theta]\right] - \mathbb{E}_{p(\theta|\mathcal{D})}\left[H[y|x, \theta]\right] \tag{8}$$

**Bayesian Query-by-Committee (B-QBC)**   Motivated by finding the optimal bias-variance trade-off, we propose a Bayesian version of the Query-by-Committee, using the MCMC samples of the hyperparameters' joint posterior. We have previously argued that the optimal bias-variance trade-off is equivalent to the optimal mode of the multimodal posterior of the hyperparameters, which is exactly what we utilize here. We use the joint posterior of the hyperparameters obtained through MCMC to draw multiple models and then query a new data point where the mean predictions $\mu_\theta(x)$ of these models disagree the most, i.e. querying the data point that maximizes the variance of $\mu_\theta(x)$. Each mean predictor $\mu_\theta(\cdot)$ drawn from the posterior is equivalent to a single model, and thus this criteria can be seen as a Bayesian variant of a Query-by-Committee, and thus denoted as Bayesian Query-by-Committee (B-QBC). Given that $\overline{\mu}(x)$ is the average mean function, B-QBC is given as

$$\text{B-QBC} = V_{p(\theta|\mathcal{D})}[\mu_\theta(x)|\theta] = \mathbb{E}_{p(\theta|\mathcal{D})}[(\mu_\theta(x) - \overline{\mu}(x))^2|\theta] \tag{9}$$

Since the models are drawn from the hyperparameters' posterior, the collection of models is dominated by models near the posterior modes. High variance in $\mu_\theta(x)$, thus corresponds to high disagreement between modes. Querying the data point that maximizes this disagreement, gives information about which mode is most likely to be the optimal one, and thus this can be seen as a mode-seeking Bayesian Query-by-Committee. To the best of our knowledge, we are the first to propose QBC based on model hypotheses drawn from the hyperparameters' joint posterior.

Table 1: Stochastic simulators used in the experiments.

| Simulator | $d$ | Noise $\sigma_\varepsilon$ | Input space | Previously used in |
|---|---|---|---|---|
| Gramacy1d | 1 | 0.1 | $[0.5, 2.5]$ | Gramacy and Lee [2012] |
| Higdon | 1 | 0.1 | $[0, 20]$ | Gramacy and Lee [2009], Gramacy [2020] |
| Gramacy2d | 2 | 0.05 | $[-2, 6]^2$ | Gramacy and Lee [2009, 2012], Sauer et al. [2022] |
| Branin | 2 | 11.32 | $[-5, 10] \times [0, 15]$ | Keane et al. [2008], Picheny et al. [2013], Cole et al. [2022] |
| Ishigami | 3 | 0.187 | $[-\pi, \pi]^3$ | Marrel et al. [2009], Cole et al. [2022] |
| Hartmann | 6 | 0.0192 | $[0, 1]^6$ | Picheny et al. [2013], Cole et al. [2022] |

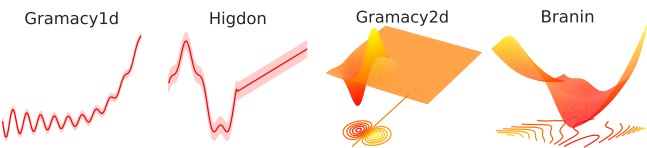

Figure 3: Visualization of the simulators (excl. Ishigami and Hartmann due to the dimensionality).

**Query by Mixture of Gaussian Processes (QB-MGP)** Bayesian Query-by-Committee (B-QBC) seek the optimal mode, but does not take the predictive performance of the model into account. Since the predictive performance, and thereby the predictive uncertainty, is also important, we extend B-QBC to consider the predictive entropy as well. We denote the new acquisition function *Query by Mixture of Gaussian Processes* (QB-MGP), because of its relation to Gaussian Mixture Models (GMMs). Using the MCMC samples, each prediction of the FBGP can be seen as an MGP, yielding the predictive posterior given as in equation (4): $\frac{1}{M} \sum_{j=1}^{M} p(\boldsymbol{y}^\star | \boldsymbol{y}, \boldsymbol{\theta}_j)$. This hierarchical predictive posterior is a mixture of $M$ Gaussians with mean $\mu_{GMM}$ and variance $\sigma^2_{GMM}$ defined as (cf. A.4):

$$\mu_{GMM}(\boldsymbol{x}) = \frac{1}{M} \sum_{j=1}^{M} \mu_{\boldsymbol{\theta}_j}(\boldsymbol{x}) \tag{10}$$

$$\sigma^2_{GMM}(\boldsymbol{x}) = \frac{1}{M} \sum_{j=1}^{M} \sigma^2_{\boldsymbol{\theta}_j}(\boldsymbol{x}) + \frac{1}{M} \sum_{j=1}^{M} (\mu_{\boldsymbol{\theta}_j}(\boldsymbol{x}) - \mu_{GMM}(\boldsymbol{x}))^2 \tag{11}$$

Finding the data point that maximizes the variance of the Mixture of Gaussian Processes (MGP) is now equivalent to simultaneously considering the B-ALM and B-QBC, i.e., the sum of the two:

$$\text{QB-MGP} = \mathbb{E}_{p(\boldsymbol{\theta}|\mathcal{D})}[\sigma^2_{\boldsymbol{\theta}}(\boldsymbol{x})|\boldsymbol{\theta}] + \mathbb{E}_{p(\boldsymbol{\theta}|\mathcal{D})}[(\mu_{\boldsymbol{\theta}}(\boldsymbol{x}) - \mu_{GMM}(\boldsymbol{x}))^2|\boldsymbol{\theta}] \tag{12}$$

Instead of using bagging, we construct the multiple GPs by using the MCMC samples of the hyperparameters' joint posterior, and then obtain a natural weighting of the GPs: the MGP will consist of more GPs with hyperparameters close to the modes than hyperparameters far away. To the best of our knowledge, we are the first to use MGP in this manner for active learning.

## 5 Experiments

In this section, we benchmark the performance of the two proposed acquisition functions against the standard acquisition functions based on the entropy, i.e., ALM, B-ALM, and BALD, on various classic simulators used in recent literature on GPs and active learning. They are all listed in Table 1, and those with less than three inputs are shown in Figure 3.[1] The multimodal posteriors of the FBGPs fitted to the simulators can be found in appendix A.5.

**Experimental settings** In the experiments, we use a zero-mean GP with an ARD RBF kernel. In each iteration of the active learning loop, the inputs are rescaled to the unit cube $[0, 1]^d$, and the outputs are standardized to have zero mean and unit variance. Following Lalchand and Rasmussen [2020], we give all the hyperparameters relatively uninformative $\mathcal{N}(0, 3)$ priors in log space. The initial data sets consist of three data points chosen by maximin Latin Hypercube Sampling [Pedregosa et al., 2011], and in each iteration, one data point is queried. The unlabeled pool $U$ consists of the

---

[1]All of them can be found at https://www.sfu.ca/~ssurjano/ [Surjanovic and Bingham, 2022].

input space discretized into 100 equidistant points along each dimension. If $U$ contains more than $10,000$ data points, we randomly sample a subset of $10,000$ data points in each iteration and use that as the new pool. The inference in FBGP is carried out using NUTS [Hoffman and Gelman, 2014] in Pyro [Bingham et al., 2019] with five chains and 500 samples, including a warm-up period with 200 samples. The remaining 1500 samples are all used for the acquisition functions. For all the predictions, we use the best mode of the hyperparameters' posterior, since the mean is of limited value when the posterior is multimodal. The best mode is computed by using a kernel density estimation with a Gaussian kernel [Pedregosa et al., 2011]. The models are implemented in GPyTorch [Gardner et al., 2018]. All experiments are repeated ten times with different initial data sets. With seven simulators and five acquisition functions, this gives 350 active learning runs, each with a running time on approximately one hour, using five CPU cores on a Threadripper 3960X. The code for reproducing the experiments is available on GitHub.[2]

**Evaluation** It is common to evaluate the performance of active learning by visually inspecting the learning or loss curves, or by measuring the performance after a specific iteration [Gramacy, 2020, Settles, 2009]. However, both procedures are inadequate to quantify how better an acquisition function is than another if we are not interested in the performance of a specific iteration but the performance in general. We are unaware of any metric for quantifying the overall performance within the regression setting, which is comparable across different data sets. Within the classification setting, Yang and Loog [2018] propose to use the area under the learning curve (AUC) based on accuracy. Since the accuracy is bounded between 0 and 1, they can compare this measure across different data sets. Likewise, O'Neill et al. [2017] compute the AUC using the root mean square error (RMSE) as the performance metric. However, since the magnitude of the RMSE is data-specific, they can not compare the performance across data sets. To resolve this problem, we suggest using the relative decrease in AUC.

We compare the relative decrease with respect to the baseline acquisition function ALM since the latter is widely used and regarded as the standard within active learning with GPs. For a metric that is lower bounded by zero, such as RMSE, the AUC will give the overall error of the acquisition function. We can directly calculate the relative decrease in error from the AUCs of the active learning acquisition functions and the baseline. If the metric has no lower bound, such as the negative log marginal likelihood (NLML), the interpretation is less intuitive. Therefore, we make a lower bound for the metric using the lowest NLML obtained across all the acquisition functions, such that the relative decrease in the AUC then can be interpreted in the same way as for the RMSE. We compare all 10 runs of each acquisition function with the 10 runs for the baseline to get a precise estimate of both the mean and standard deviation of the relative decreases. Since the relative decrease is a ratio, we compute the unbiased estimates using the formulas from Van Kempen and Van Vliet [2000]. See Appendix A.6 for the formulas and the pseudo-code.

## 5.1 Experiments with 6 simulators

We benchmark the acquisition functions on the six classic simulators. We divide the simulators into subgroups and describe how the functions work in different complexities and with multiple inputs, effectively encompassing three distinct modeling scenarios. The following paragraphs describe the active learning curves in Figure 4.

**Noise or signal** The simulator *Gramacy1d* has previously been used to study the effect of the noise-term in GPs [Gramacy and Lee, 2012]. The simulator has a periodic signal that is hard to reveal if the data points are not queried cleverly. Both B-QBC and QB-MGP reach convergence simultaneously, while the other acquisition functions struggle in distinguishing noise from the signal.

**Linear and non-linear output regions** The two simulators *Higdon* and *Gramacy2d* have been used to illustrate cases where GPs struggle in modeling the data due to the output signal having both linear and non-linear regions [Gramacy and Lee, 2009]. Our experiments on these simulators show the performance of the acquisition functions when the GP is an non-optimal choice of model. Querying data points in the linear and non-linear regions will yield a GP with a longer and shorter length scale, respectively. For Higdon, both B-QBC and QB-MGP balance the sampling since the corresponding NLML and RMSE are low, likewise, for Gramacy2d, QB-MGP is achieving both the

---

[2]https://github.com/coriis/active-learning-fbgp

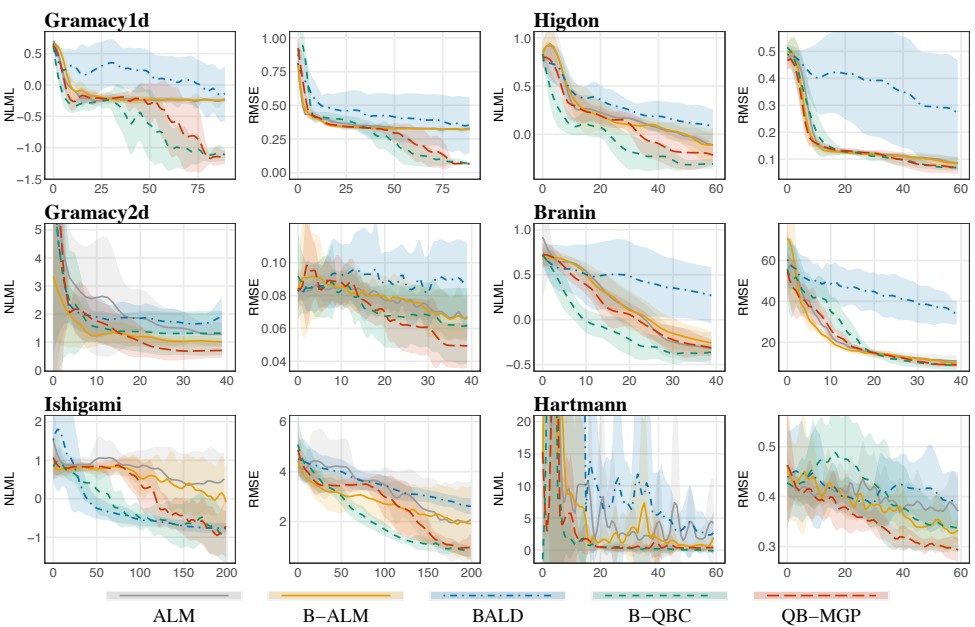

Figure 4: Performance across the 10 runs ±1 standard deviation. The $x$-axis represents the number of iterations.

Table 2: The relative decrease in the area under the active learning curves compared to the active learning curve of the baseline acquisition function ALM.

| | GRAMACY1D | HIGDON | GRAMACY2D | BRANIN | ISHIGAMI | HARTMANN | OVERALL PERFORMANCE | |
|---|---|---|---|---|---|---|---|---|
| DIMENSIONS | 1 | 1 | 2 | 2 | 3 | 6 | MEAN | MEDIAN |
| NLML: RELATIVE DECREASE IN AUC (%) | | | | | | | | |
| ALM | 0.0 ±0.6 | 0.0 ±1.2 | 0.0 ±6.4 | 0.0 ±1.2 | 0.0 ±1.1 | 0.0 ±22.2 | 0.0 ±0.0 | 0.0 ±0.0 |
| B-ALM | -1.7 ±0.5 | 1.2 ±1.4 | **43.2 ±3.1** | -2.4 ±1.6 | 7.9 ±1.3 | 48.2 ±10.4 | 16.1 ±21.3 | 4.6 ±21.3 |
| BALD | -28.1 ±2.1 | -13.6 ±2.0 | 7.8 ±5.0 | -36.0 ±3.6 | **39.1 ±1.1** | -17.3 ±22.0 | -8.0 ±25.1 | -15.5 ±25.1 |
| B-QBC | **36.3 ±1.0** | **41.5 ±1.1** | 23.9 ±4.6 | **33.1 ±1.2** | 37.5 ±0.8 | 75.5 ±5.1 | **41.3 ±16.2** | **36.9 ±16.2** |
| QB-MGP | 20.6 ±0.9 | 15.9 ±1.4 | 30.6 ±4.4 | 7.1 ±1.1 | 21.7 ±0.7 | **80.1 ±3.9** | 29.3 ±23.8 | 21.1 ±23.8 |
| RMSE: RELATIVE DECREASE IN AUC (%) | | | | | | | MEAN | MEDIAN |
| ALM | 0.0 ±0.2 | 0.0 ±0.7 | 0.0 ±0.5 | 0.0 ±1.0 | 0.0 ±3.0 | 0.0 ±0.7 | 0.0 ±0.0 | 0.0 ±0.0 |
| B-ALM | 2.4 ±0.2 | 5.0 ±0.9 | -1.0 ±1.0 | **4.3 ±1.1** | 15.7 ±2.7 | 3.8 ±0.8 | 5.0 ±5.1 | 4.0 ±5.1 |
| BALD | -23.3 ±3.9 | -133.0 ±8.8 | -14.2 ±2.1 | -111.3 ±3.9 | -6.1 ±2.5 | -3.7 ±0.9 | -48.6 ±52.8 | -18.7 ±52.8 |
| B-QBC | **20.8 ±0.9** | 4.5 ±1.0 | 5.9 ±1.2 | -8.6 ±1.6 | **36.4 ±1.4** | -2.7 ±1.4 | 9.2 ±15.1 | 4.7 ±15.1 |
| QB-MGP | 19.6 ±0.8 | **10.4 ±0.8** | **8.8 ±0.7** | 3.1 ±1.0 | 18.7 ±2.0 | **11.7 ±0.6** | **12.0 ±5.7** | **11.1 ±5.7** |

lowest NLML and RMSE. Overall, these results show that when the GP is inadequate to model the data, both B-QBC and QB-MGP perform better than the other acquisition functions.

**Multiple inputs** To evaluate the performance on higher dimensions, we consider the smooth 2d *Branin* simulator, the strongly non-linear 3d *Ishigami* simulator, and the 6d *Hartmann* simulator with six local minima. BALD underperforms on Branin, but the other acquisition functions have similar performance, with B-QBC having an overall better NLML. For Ishigami, BALD, B-QBC, and QB-MGP reach the best NLML, but the earlier iterations show that BALD is slightly better than the other two acquisition functions. For Hartmann, B-QBC and QB-MGP are the most stable and best in terms of the NLML, whereas the latter achieves the lowest RMSE.

## 5.2 The Overall Performance

From the visual inspection of Figure 4, it is hard to measure how good each of the acquisition functions are. In Table 2, we quantify the performance using the method described earlier, based on the relative decrease in the area under the curve (AUC).

First of all, it is clear that there is not a single acquisition function that is performing the best for all the simulators. B-QBC achieves the largest decrease in AUC for either NLML or RMSE in five cases, and QB-MGP is the next best, having the largest decrease four times. This is reflected in the overall performance, where B-QBC and QB-MGP are the best performing acquisition functions in terms of the marginal likelihood and root mean square error, respectively. Oppositely, BALD is the worst performing acquisition function overall, not even better than the baseline, which suggests that this acquisition function might not be suited for Gaussian Processes within the regression setting. Given the overall good consistent performance of B-QBC and QB-MGP (B-QBC only worse than the baseline twice), we say that they are robust to different complexities in the simulators' outputs.

## 5.3 Limitations

Gaussian Processes are known to be computationally expensive [Williams and Rasmussen, 2006]. The computational cost scales cubically with the number of data points in the data set, i.e., $O(n^3)$, and the GPs are thus only suited for small data sets. A fully Bayesian GP is even more computationally expensive because of the MCMC sampling. There exist methods to circumvent this, e.g, variational inference (VI), but often at the cost of the approximation of the joint posterior of the hyperparameters [Lalchand and Rasmussen, 2020]. For future work, we will investigate if the posterior can be approximated by VI, using several initializations, or if it is possible to achieve similar performance by alternating between using a fully Bayesian GP and a regular GP.

This paper is based on empirical results that are dependent on specific simulators. The simulators represent diverse and distinct classic homoscedastic simulators and are representative of the engineering problems occurring in the real world [Gramacy, 2020, Sauer et al., 2022, Cole et al., 2022]. However, an interesting case that is less often investigated in the literature on active learning with simulators, is a simulator with heteroscedastic noise.

A common heteroscedastic case study is the motorcycle data set [Silverman, 1985, Gramacy and Lee, 2008, Gramacy, 2020]. We create a corresponding simulator by fitting a variational GP [Hensman et al., 2015] to the motorcycle accident data. For reproducibility, the mean and standard deviation of the simulator is given in appendix (A.7). The experiments on the *Motorcycle* simulator explore how the active learning acquisition functions perform when the simulator has heteroscedastic noise, but we model it with a homoscedastic GP. The simulator and the results are seen in Figure 5. Most conspicuous is the poor and good performance of B-QBC and QB-MGP, respectively. In appendix A.8, we show that B-QBC is misled by the heteroscedastic noise and focuses too much on the disagreement in the middle, and that the B-ALM component of QB-MGP acts as a diversity measure that encourages more exploration since it aims in reducing the overall predictive uncertainty.

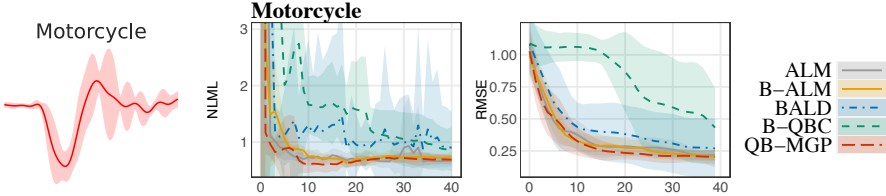

Figure 5: Left: Visualization of the Motorcycle simulator. Right: Performance across 10 runs $\pm 1$ standard deviation. The $x$-axis represent the number of iterations.

Another aspect of the work in this paper is the use of the domain and expert knowledge. The incorporation of the domain and expert guidance regarding the simulator under study can be a decisive factor in a successful active learning strategy. However, in many practical situations, such a priori domain expertise may not be readily accessible or even translatable into the functional structure of the model as useful modeling information. On these occasions, generic tools that are robust enough to handle a plethora of diverse simulation output behaviors are prudently advisable. If we had such information regarding the functional complexity, e.g., knowing that the signal is periodic, this study does not show which of the acquisition functions is best. In future work, it would indeed be interesting to see if B-QBC and QB-MGP would perform equally well.

# 6 Conclusion

In this paper, we propose two active learning acquisition functions: Bayesian Query-by-Committee (B-QBC) and Query by a Mixture of Gaussian Processes (QB-MGP), both of which are suited for fully Bayesian GPs. They are designed to explicitly handle the well-known bias-variance trade-off by optimization of the GP's two hyperparameters, length scale and noise-term. We empirically show that they query new data points more efficiently than previously used acquisition functions. Across six classic simulators, which cover different complexities and numbers of inputs, we show that B-QBC and QB-MGP are the two functions that achieve the best marginal likelihood and root mean square error, respectively, with the fewest iterations. On average, across the simulators, B-QCB reduced the negative marginal log-likelihood with $41\%$, and QB-MGP decreased the root mean square error with $12\%$ compared to the baseline. To this end, we believe that the proposed acquisition functions are robust enough to handle a variety of diverse simulation output behaviors, while being entirely independent of any prior understanding of the underlying output distributions of the simulator.

## Acknowledgments and Disclosure of Funding

This work was supported by NOSTROMO, framed in the scope of the SESAR 2020 Exploratory Research topic SESAR-ER4-26-2019, funded by SESAR Joint Undertakingthrough the European Union's Horizon 2020 research and innovation programme under grant agreement No 892517.

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
