## Supplemental Material for "Bayesian Active Learning with Fully Bayesian Gaussian Processes"

## A   Appendix

### A.1   Gaussian Process

The generative model of a GP is given as

$$\boldsymbol{f}|X \sim \mathcal{N}(0, K_{\boldsymbol{\theta}}) \quad \text{and} \quad \boldsymbol{y}|\boldsymbol{f} \sim \mathcal{N}(\boldsymbol{f}, \sigma_{\varepsilon}^2 \mathbb{I}) \tag{13}$$

which defines the prior over the latent functions and the likelihood of the data, respectively. The generative model implies that the joint posterior over the latent parameters given as

$$p(\boldsymbol{f}|\boldsymbol{y}, X) = \frac{p(\boldsymbol{y}|\boldsymbol{f})p(\boldsymbol{f}|X)}{p(\boldsymbol{y}|X)}. \tag{14}$$

Using the marginalization property for Gaussian distributions, the marginal distribution of $\boldsymbol{y}$ is given by

$$p(\boldsymbol{y}|X) = \int p(\boldsymbol{y}|\boldsymbol{f})p(\boldsymbol{f}|X)d\boldsymbol{f} = \mathcal{N}(\boldsymbol{y}|0, K_{\boldsymbol{\theta}}). \tag{15}$$

Thus, the hyperparameters $\boldsymbol{\theta}$ of the kernel can be optimized by the maximum marginal likelihood - or if we add priors on the $\boldsymbol{\theta}$ - by maximum a posteriori:

$$\hat{\boldsymbol{\theta}} = \arg \max_{\boldsymbol{\theta}} \left( \log \mathcal{N}(\boldsymbol{y}|\boldsymbol{0}, \boldsymbol{\theta}) + \log p(\boldsymbol{\theta}) \right). \tag{16}$$

The log marginal likelihood is given as [Williams and Rasmussen, 2006]:

$$\log p(\boldsymbol{y}|X) = -\frac{1}{2}\boldsymbol{y}^{\top} K_{\boldsymbol{\theta}}^{-1} \boldsymbol{y} - \frac{1}{2} \log |K_{\boldsymbol{\theta}}| - \frac{n}{2} \log 2\pi \tag{17}$$

### A.2   Derivations of the entropy for a Gaussian distribution

The following derivation shows that the entropy of a Gaussian distribution $\mathcal{N}(\mu, \sigma^2)$ is proportional to the variance $\sigma^2$.

$$H[X] = \mathbb{E}\left[-\ln p(X)\right] \tag{18}$$

$$= \mathbb{E}\left[\frac{1}{2} \ln 2\pi\sigma^2 + \frac{(X-\mu)^2}{2\sigma^2}\right] \tag{19}$$

$$= \frac{1}{2} \ln 2\pi\sigma^2 + \frac{1}{2\sigma^2}\mathbb{E}\left[(X-\mu)^2\right] \tag{20}$$

$$= \frac{1}{2} \ln 2\pi\sigma^2 + \frac{\sigma^2}{2\sigma^2} \tag{21}$$

$$= \frac{1}{2} \ln 2\pi e\sigma^2, \quad \text{since } \frac{1}{2} = \frac{1}{2} \ln e \tag{22}$$

$$\propto \sigma^2 \tag{23}$$

### A.3   Derivations of BALD

BALD is given as [Houlsby et al., 2011]:

$$I[\hat{\boldsymbol{\theta}}, y|\boldsymbol{x}, \mathcal{D}] = H[\hat{\boldsymbol{\theta}}|\mathcal{D}] - \mathbb{E}_{p(y|\boldsymbol{x}, \mathcal{D})}[H[\hat{\boldsymbol{\theta}}|y, \boldsymbol{x}, \mathcal{D}]] \tag{24}$$

$$= H[y|\boldsymbol{x}, \mathcal{D}] - \mathbb{E}_{p(\hat{\boldsymbol{\theta}}|\mathcal{D})}[H[y|\boldsymbol{x}, \hat{\boldsymbol{\theta}}]] \tag{25}$$

For a regular GP with a homoscedastic Gaussian likelihood ($\mathcal{N}(0, \sigma_{\varepsilon}^2)$), BALD is equivalent to ALM

$$I[\hat{\boldsymbol{\theta}}, y|\boldsymbol{x}, \mathcal{D}] = H[y|\boldsymbol{x}, \mathcal{D}] - \mathbb{E}_{p(\hat{\boldsymbol{\theta}}|\mathcal{D})}[H[y|\boldsymbol{x}, \hat{\boldsymbol{\theta}}]] \tag{26}$$

$$= H[y|\boldsymbol{x}, \mathcal{D}] - \mathbb{E}_{p(\boldsymbol{f}|\mathcal{D})}[H[y|\boldsymbol{x}, \boldsymbol{f}]] \tag{27}$$

$$= \frac{1}{2} \ln(2\pi\sigma^2(\boldsymbol{x})) - \sigma_{\varepsilon}^2 \tag{28}$$

$$\propto \sigma^2(\boldsymbol{x}) - const. \tag{29}$$

### A.3.1 Fully Bayesian GP

For a Fully Bayesian GP (FBGP) with a homoscedastic Gaussian likelihood ($\mathcal{N}(0, \sigma_\varepsilon^2|\boldsymbol{\theta})$), there are three different ways to handle the hyperparameters Houlsby et al. [2011]:

1. The parameter of interest is $\boldsymbol{f}$, and the hyperparameters $\boldsymbol{\theta}$ are nuisance parameters,
2. The parameters of interest are both $\boldsymbol{f}$ and the hyperparameters $\boldsymbol{\theta}$,
3. The parameters of interest are the hyperparameters $\boldsymbol{\theta}$, and the parameter $\boldsymbol{f}$ is a nuisance parameter.

In the first and the second case, BALD is equivalent to B-ALM (here derived for case 1):

$$I[\boldsymbol{f}, \boldsymbol{\theta}, y|\boldsymbol{x}, \mathcal{D}] = H\left[\mathbb{E}_{p(\boldsymbol{f},\boldsymbol{\theta}|D)}[y|\boldsymbol{x}, \mathcal{D}, \boldsymbol{f}, \boldsymbol{\theta}]\right] - \mathbb{E}_{p(\boldsymbol{f}|\mathcal{D})}\left[H[\mathbb{E}_{p(\boldsymbol{\theta}|f,\mathcal{D})}[y|\boldsymbol{x}, \boldsymbol{f}, \boldsymbol{\theta}]]\right] \tag{30}$$

$$= H\left[\mathbb{E}_{p(\boldsymbol{\theta}|\mathcal{D})}[y|\boldsymbol{x}, \boldsymbol{\theta}]\right] - \mathbb{E}_{p(\boldsymbol{f}|\mathcal{D})}\left[H[\mathbb{E}_{p(\boldsymbol{\theta}|f,\mathcal{D})}[y|\boldsymbol{x}, \boldsymbol{f}, \boldsymbol{\theta}]]\right] \tag{31}$$

For the first term in the equation above, $\boldsymbol{f}$ is integrated out such that the predictive posterior $p(y|\boldsymbol{x})$ now corresponds to the regular GP predictive posterior. Since the $\boldsymbol{f}$ is given by the outer expectation and $y$ is dependent on $\boldsymbol{f}$ in the inner expectation, the second term is independent of $\boldsymbol{x}$. Hence, BALD is then given as:

$$I[\boldsymbol{f}, \boldsymbol{\theta}, y|\boldsymbol{x}, \mathcal{D}] \propto \mathbb{E}_{p(\boldsymbol{\theta}|\mathcal{D})}[\sigma_{\boldsymbol{\theta}}^2(\boldsymbol{x})|\boldsymbol{\theta}] - \mathbb{E}_{p(\boldsymbol{\theta}|\mathcal{D})}[\sigma_\varepsilon^2|\boldsymbol{\theta}]. \tag{32}$$

$$\propto \mathbb{E}_{p(\boldsymbol{\theta}|\mathcal{D})}[\sigma_{\boldsymbol{\theta}}^2(\boldsymbol{x})|\boldsymbol{\theta}] - const. \tag{33}$$

In the third case, where the hyperparameters are the parameters of interest, BALD is given as

$$I[\boldsymbol{f}, \boldsymbol{\theta}, y|\boldsymbol{x}, \mathcal{D}] = H\left[\mathbb{E}_{p(\boldsymbol{f},\boldsymbol{\theta}|\mathcal{D})}[y|\boldsymbol{x}, \mathcal{D}, \boldsymbol{f}, \boldsymbol{\theta}]\right] - \mathbb{E}_{p(\boldsymbol{\theta}|\mathcal{D})}\left[H[\mathbb{E}_{p(\boldsymbol{f}|\boldsymbol{\theta},\mathcal{D})}[y|\boldsymbol{x}, \boldsymbol{f}, \boldsymbol{\theta}]]\right] \tag{34}$$

The expectation in the second term is now equal to the standard GP predictive posterior, and thus BALD is given as:

$$I[\boldsymbol{f}, \boldsymbol{\theta}, y|\boldsymbol{x}, \mathcal{D}] = H\left[\mathbb{E}_{p(\boldsymbol{\theta}|\mathcal{D})}[y|\boldsymbol{x}, \mathcal{D}, \boldsymbol{\theta}]\right] - \mathbb{E}_{p(\boldsymbol{\theta}|\mathcal{D})}\left[H[y|\boldsymbol{x}, \boldsymbol{\theta}]\right] \tag{35}$$

## A.4 Proof of moments for Gaussian Mixture Model

Using the MCMC samples, each prediction of the FBGP can be seen as a GMM, yielding the predictive posterior given as

$$p(y|\boldsymbol{x}, \mathcal{D}) = \int p(y|\boldsymbol{x}, \mathcal{D}, \boldsymbol{\theta})\, p(\boldsymbol{\theta}|y)d\boldsymbol{\theta} \quad \simeq \quad \frac{1}{M}\sum_{j=1}^{M} p(y|\boldsymbol{x}, \mathcal{D}, \boldsymbol{\theta}_j), \quad \boldsymbol{\theta}_j \sim p(\boldsymbol{\theta}|\mathcal{D}) \tag{36}$$

This hierarchical predictive posterior is a mixture of $M$ Gaussians with mean $\mu_{GMM}$ and variance matrix $\sigma_{GMM}^2$. The mean $\mu_{GMM}$ derived as:

$$\mu_{GMM}(y) = \mathbb{E}[y|\boldsymbol{x}] \tag{37}$$

$$= \mathbb{E}\left[\frac{1}{M}\sum_{j=1}^{M} p(y|\boldsymbol{x}, \boldsymbol{\theta}_j)\right] \tag{38}$$

$$= \mathbb{E}\left[\frac{1}{M}\sum_{j=1}^{M} \mathcal{N}(y|\mu_{\boldsymbol{\theta}_j}(\boldsymbol{x}), \sigma_{\boldsymbol{\theta}_j}^2(\boldsymbol{x}))\right] \tag{39}$$

$$= \frac{1}{M}\sum_{j=1}^{M} \mathbb{E}\left[\mathcal{N}(y|\mu_{\boldsymbol{\theta}_j}(\boldsymbol{x}), \sigma_{\boldsymbol{\theta}_j}^2(\boldsymbol{x}))\right] \tag{40}$$

$$= \frac{1}{M}\sum_{j=1}^{M} \mu_{\boldsymbol{\theta}_j}(\boldsymbol{x}) \tag{41}$$

The variance $\sigma_{GMM}^2$ is derived as:

$$\sigma_{GMM}^2(y) = \mathbb{E}\left[\,||y - \mathbb{E}[y|\boldsymbol{x}]||^2 \,|\, \boldsymbol{x}\right] \tag{42}$$

For short, we set $\bar{y} := \mu_{GMM}(y) = \mathbb{E}[y|\boldsymbol{x}]$

$$= \mathbb{E}\left[\,||y - \bar{y}||^2\,|\,\boldsymbol{x}\right] \tag{43}$$

$$= \int ||y - \bar{y}||^2 p(y|\boldsymbol{x})\,dy \tag{44}$$

$$= \int ||y - \bar{y}||^2 \frac{1}{M}\sum_{j=1}^{M} p\left(y|\boldsymbol{x}, \boldsymbol{\theta}_j\right)\,dy \tag{45}$$

$$= \frac{1}{M}\sum_{j=1}^{M} \int ||y - \bar{y}||^2 p\left(y|\boldsymbol{x}, \boldsymbol{\theta}_j\right)\,dy \tag{46}$$

$$= \frac{1}{M}\sum_{j=1}^{M} \int \left(y^2 + \bar{y}^2 - 2y\bar{y}\right)\mathcal{N}\left(y|\mu_{\boldsymbol{\theta}_j}(\boldsymbol{x}), \sigma_{\boldsymbol{\theta}_j}(\boldsymbol{x})\right)\,dy \tag{47}$$

$$= \frac{1}{M}\sum_{j=1}^{M} \mathbb{E}_{\mathcal{N}\left(y|\mu_{\boldsymbol{\theta}_j}(\boldsymbol{x}), \sigma_{\boldsymbol{\theta}_j}(\boldsymbol{x})\right)}\left[y^2 + \bar{y}^2 - 2y\bar{y}\right] \tag{48}$$

If we set $\mathcal{N}_{\boldsymbol{\theta}} := \mathcal{N}\left(y|\mu_{\boldsymbol{\theta}_j}(\boldsymbol{x}), \sigma_{\boldsymbol{\theta}_j}(\boldsymbol{x})\right)$, we can re-write it as:

$$= \frac{1}{M}\sum_{j=1}^{M} \mathbb{E}_{\mathcal{N}_{\boldsymbol{\theta}}}[y^2] + \bar{y}^2 + 2\bar{y}\,\mathbb{E}_{\mathcal{N}_{\boldsymbol{\theta}}}[y] \tag{49}$$

Using that $V[X] = \mathbb{E}[X^2] - \mathbb{E}[X]^2$, we get

$$= \frac{1}{M}\sum_{j=1}^{M} \left(V_{\mathcal{N}_{\boldsymbol{\theta}}}[y] + \mathbb{E}_{\mathcal{N}_{\boldsymbol{\theta}}}[y]^2 + \bar{y}^2 - 2\bar{y}\,\mathbb{E}_{\mathcal{N}_{\boldsymbol{\theta}}}[y]\right) \tag{50}$$

$$= \frac{1}{M}\sum_{j=1}^{M} \left(\sigma_{\boldsymbol{\theta}_j}^2(\boldsymbol{x}) + \mu_{\boldsymbol{\theta}_j}(\boldsymbol{x})^2 + \mu_{GMM}(\boldsymbol{x})^2 - 2\mu_{GMM}(\boldsymbol{x})\mu_{\boldsymbol{\theta}_j}(\boldsymbol{x})\right) \tag{51}$$

$$= \frac{1}{M}\sum_{j=1}^{M} \left(\sigma_{\boldsymbol{\theta}_j}^2(\boldsymbol{x}) + ||\mu_{\boldsymbol{\theta}_j}(\boldsymbol{x}) - \mu_{GMM}(\boldsymbol{x})||^2\right) \tag{52}$$

## A.5 Posterior of the hyperparameters

For the one dimensional simulators, the resulting FBGP has two hyperparameters: the length scale and noise-term. For the $d$-dimensional simulators, an FBGP with an ARD RBF kernel has $d + 1$ hyperparameters, and thus we cannot plot the joint posterior for simulators with more than one input dimension. Instead we plot the joint posterior of the hyperparameter of an FBGP with an RBF kernel. The following posteriors are all averaged across the 10 experiments for a particular data size. In Figure 6, the five posteriors that gave a multimodal posterior using the RBF kernel are shown.

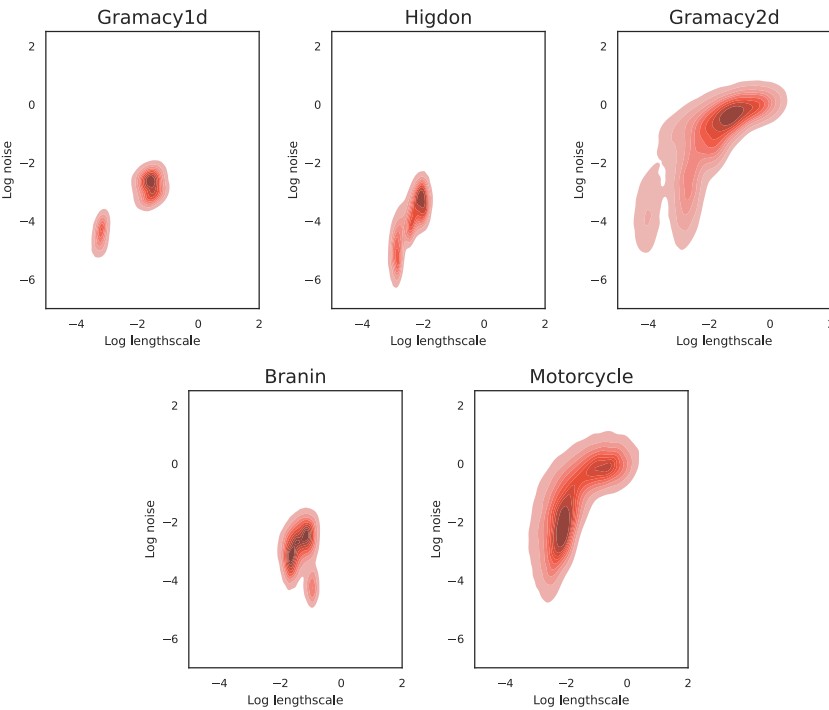

Figure 6: The joint posterior of the hyperparameters of a RBF GP fitted to data from the simulators. The data sizes are for Gramacy1d: 70, Higdon: 40, Gramacy2d: 40, Branin: 40, and Motorcycle: 10.

For the 3d simulator Ishigami and 6d simulator Hartmann, the RBF kernel gave only one mode using this visualization, see Figure 7. Due to the dimensionality, it is likely that with an ARD RBF kernel that the posterior will have more than one mode.

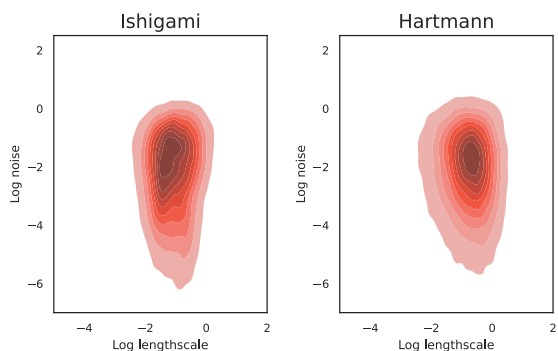

Figure 7: The joint posterior of the hyperparameters of a RBF GP fitted to 40 data points from the simulators.

## A.6 Mean and variance for the relative decrease in area under the loss curve

To compute the mean and variance of the relative decrease in area under the learning curve (RD-AUC), we use the unbiased estimates of the mean and variance of the ratios. Let $\text{AUC}_b$ and $\text{AUC}_{new}$ be the AUCs for the baseline and new acquisition function, respectively, then the RD-AUC is

$$\text{RD-AUC} = \frac{\text{AUC}_{\text{new}} - \text{AUC}_{\text{b}}}{\text{AUC}_{\text{b}}} \tag{53}$$

Note that if the AUC is computed by a metric that is not lower bounded, e.g., the negative marginal likelihood, all the AUCs should be subtracted by the best obtained performance across all the tested acquisition functions, such that we create an artificial lower bound.

Let $\mu_n$ and $\mu_d$ be the expectation of the nominator and denominator, respectively, then the unbiased estimate of the expectation of RD-AUC is (cf. Van Kempen and Van Vliet [2000] eq. (9))

$$\mu_{\text{RD-AUC}} = \frac{\mu_n}{\mu_d} \tag{54}$$

Let $R$ be the number of times the acquisition function is run with different initial data sets, and let $\sigma_n^2$, $\sigma_d^2$, and $\sigma_{nd}^2$ be the variance of and covariance between the nominator and denominator. Then the unbiased estimate of the variance of RD-AUC is (cf. Van Kempen and Van Vliet [2000] eq. (10))

$$\sigma_{\text{RD-AUC}}^2 = \frac{1}{R}\left(\frac{\sigma_n^2}{\mu_d^2} + \frac{\mu_n^2\sigma_d^2}{\mu_d^4} - \frac{2\mu_n\sigma_{nd}^2}{\mu_d^3}\right) \tag{55}$$

The pseudo-code to compute the mean and variance of the RD-AUC is given in Algorithm 1.

---

**Algorithm 1** Mean and variance for the relative decrease in area under the loss curve (AUC)

---

**Require:** Let $\text{AUC}_b$ and $\text{AUC}_{new}$ be the AUCs for the baseline and new acquisition function, respectively. Apply the acquisition function $R$ times with different initial data sets to get the two sets $\{\text{AUC}_b^r\}_{r=1}^R$ and $\{\text{AUC}_{new}^r\}_{r=1}^R$.

1: **procedure** RD-AUC      ▷ Relative Decrease in AUC
2:    **if** AUC is lower bounded **then**
3:      $\text{AUC}_{best} \leftarrow$ lower bound      ▷ e.g., for RMSE it would be 0
4:    **else**
5:      $\text{AUC}_{best} \leftarrow$ best performance across all the tested acquisition functions
6:    **end if**
7:    n, d $\leftarrow [\,], [\,]$      ▷ Nominator (n) and denominator (d)
8:    **for** $\text{AUC}_b \in \{\text{AUC}_b^r\}_{r=1}^R$ **do**
9:      **for** $\text{AUC}_{new} \in \{\text{AUC}_{new}^r\}_{r=1}^R$ **do**
10:        n.append($\text{AUC}_b - \text{AUC}_{new}$)
11:        d.append($\text{AUC}_b - \text{AUC}_{best}$)
12:      **end for**
13:    **end for**
14:    $\mu_n \leftarrow \frac{1}{R}\sum_{r=1}^R \text{n}_r$      ▷ Compute mean, variance, and covariance
15:    $\mu_d \leftarrow \frac{1}{R}\sum_{r=1}^R \text{d}_r$
16:    $\sigma_n^2, \sigma_d^2, \sigma_{nd}^2 \leftarrow V[\text{n}], V[\text{d}], Cov(\text{n}, \text{d})$
17:    $\mu_{\text{RD-AUC}} \leftarrow \mu_n/\mu_d$      ▷ Expectation of RD-AUC
18:    $\sigma_{\text{RD-AUC}}^2 \leftarrow \frac{1}{R}\left(\frac{\sigma_n^2}{\mu_d^2} + \frac{\mu_n^2\sigma_d^2}{\mu_d^4} - \frac{2\mu_n\sigma_{nd}^2}{\mu_d^3}\right)$      ▷ Variance of RD-AUC
19:    **return** $\mu_{\text{RD-AUC}}, \sigma_{\text{RD-AUC}}^2$
20: **end procedure**

---

### A.7 Motorcycle simulator

The motorcycle simulator is created by fitting a variational GP [Hensman et al., 2015] to the motorcycle accident data [Silverman, 1985], where the mean and standard deviation then is extracted. The simulator is fully specified by the following mean standard deviation.

```
mean = [0.5107, 0.5032, 0.4939, 0.4843, 0.4765, 0.4718, 0.4713,
        0.4748, 0.4805, 0.4856, 0.4870, 0.4824, 0.4724, 0.4604,
        0.4523, 0.4543, 0.4695, 0.4953, 0.5216, 0.5319, 0.5067,
        0.4284, 0.2868, 0.0825,-0.1722,-0.4559,-0.7433,-1.0112,
       -1.2435,-1.4339,-1.5851,-1.7049,-1.8015,-1.8788,-1.9333,
       -1.9549,-1.9298,-1.8453,-1.6947,-1.4804,-1.2140,-0.9140,
       -0.6013,-0.2950,-0.0080, 0.2536, 0.4900, 0.7046, 0.9006,
        1.0778, 1.2317, 1.3542, 1.4362, 1.4710, 1.4573, 1.4012,
        1.3147, 1.2139, 1.1143, 1.0277, 0.9589, 0.9056, 0.8605,
        0.8146, 0.7605, 0.6959, 0.6241, 0.5533, 0.4935, 0.4538,
        0.4395, 0.4501, 0.4795, 0.5173, 0.5507, 0.5678, 0.5600,
        0.5241, 0.4632, 0.3869, 0.3092, 0.2456, 0.2098, 0.2103,
        0.2482, 0.3164, 0.4015, 0.4868, 0.5568, 0.6006, 0.6147,
        0.6033, 0.5769, 0.5487, 0.5311, 0.5324, 0.5549, 0.5950,
        0.6441, 0.6919, 0.7285]

stddev = [0.0477, 0.0400, 0.0463, 0.0514, 0.0529, 0.0502, 0.0438,
          0.0366, 0.0330, 0.0328, 0.0327, 0.0318, 0.0315, 0.0331,
          0.0363, 0.0433, 0.0560, 0.0702, 0.0780, 0.0733, 0.0615,
          0.0799, 0.1468, 0.2393, 0.3408, 0.4382, 0.5196, 0.5747,
          0.5959, 0.5800, 0.5295, 0.4540, 0.3712, 0.3067, 0.2822,
          0.2915, 0.3074, 0.3135, 0.3156, 0.3360, 0.3899, 0.4658,
          0.5385, 0.5873, 0.6045, 0.5975, 0.5860, 0.5901, 0.6133,
          0.6377, 0.6380, 0.5951, 0.5022, 0.3669, 0.2227, 0.1980,
          0.3483, 0.5319, 0.6956, 0.8186, 0.8886, 0.8987, 0.8470,
          0.7373, 0.5789, 0.3876, 0.1898, 0.1067, 0.2472, 0.3799,
          0.4620, 0.4847, 0.4492, 0.3643, 0.2463, 0.1253, 0.1068,
          0.2006, 0.2817, 0.3230, 0.3195, 0.2769, 0.2092, 0.1377,
          0.0911, 0.0856, 0.0919, 0.0940, 0.1116, 0.1568, 0.2117,
          0.2548, 0.2730, 0.2623, 0.2282, 0.1857, 0.1545, 0.1445,
          0.1445, 0.1437, 0.1542]
```

### A.8    Motorcycle simulator: B-QBC vs. QB-MGP

In Figure 8, the data points queried by B-QBC and QB-MGP are compared for two runs after 20 and 40 iterations. The behavior of the two runs are consistent with the other runs. After 20 iterations, B-QBC has queried many points around the middle, because of the disagreement between the modes, cf. Figure 6. Oppositely, the B-ALM component of the QB-MGP seems to encourage exploration, and thus QB-MGP does not get stuck exploring the two modes. After 40 iterations, both acquisition functions have explored the space similarly.

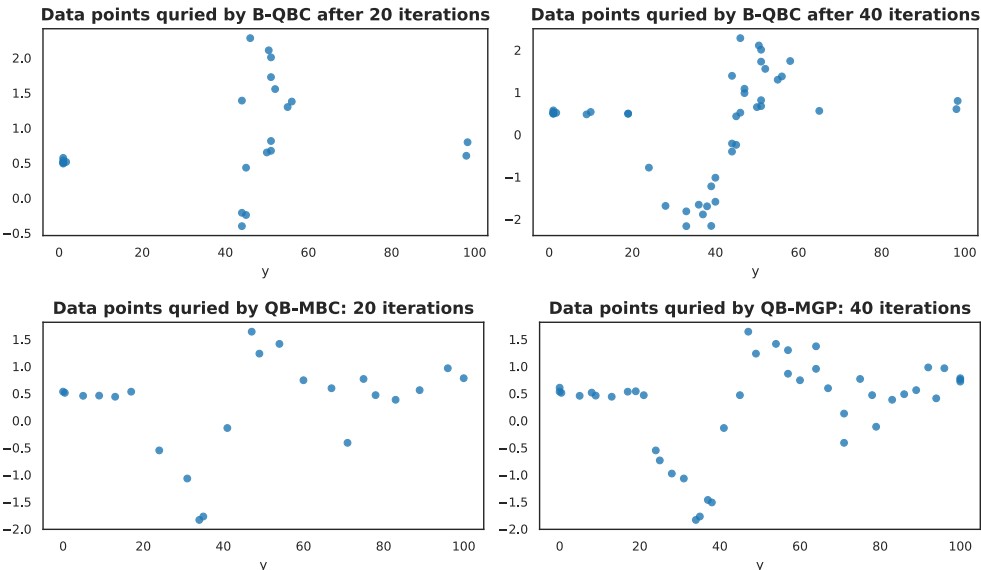

Figure 8: Comparison of the data acquisition of B-QBC and QB-MGP for the motorcycle simulator.