# OpenReview forum: "Bayesian Active Learning with Fully Bayesian Gaussian Processes"
_NeurIPS.cc/2022/Conference — NeurIPS 2022 Accept_

### Official Review · Reviewer_8dtu · 2022-07-09

**Rating:** 7
**Confidence:** 4
**Soundness:** 3 good
**Presentation:** 4 excellent
**Contribution:** 3 good

**Summary:**

This work proposes the use of fully Bayesian GPs in the context of Bayes opt and active learning. They propose fully Bayesian versions of known acquisition functions that leverage information from the hyperparameter posterior and also suggest using the posterior predictive variance of the  GMM (non-Gaussian) process to derive an acquisition function. The claim is that the Bayesian variants yield more effective acquisition functions --  implicitly because they approximate the posterior predictive uncertainty better than ML-II. They demonstrate and analyse the performance across methods on 6 simulator tasks

**Questions:**

1) Isn't the most reasonable mode given by the mode where the marginal likelihood is maximized? and typically, the modes manifest in very starkly different marginal likelihood values.
2) The underperformance of B-QBC on the heteroscedastic motorcycle data is not sufficiently explained, QB-MGP performs best but in terms of the criteria is not so different than B-QBC.
3) Both B-QBC and QB-MGP use hyper samples from the posterior - then \mu_GMM is the same as the mean of the models corresponding to thetas drawn from the hyper posterior \bar{\mu}? Their performance should be very similar but they are quite different - is there more insight on this?

**Limitations:**

The authors do discuss limitations - but conspicuously miss out the obvious one which is the compute for FBGPs.

**Strengths And Weaknesses:**

Strengths:
* Interesting interpretation of the multi-modal marginal likelihood surface in terms of the bias-variance tradeoff parlance.
* Good presentation, well-written and clear.
* Decent experimental evaluation.
* Approximating the hyperparameter posterior - a relatively understudied area in literature.

Weakness:

*A major concern is the computational cost of running FBGP at each iteration. A naive strategy (albeit sufficient for small datasets like the ones studied) would scale as O(MN^{3}) where M is the number of samples - how exactly can this method scale? perhaps by interleaving FBGP at intervals with traditional ML-II optimisation - a more sophisticated strategy would be required.

---

> ### Author Response · Authors · 2022-08-02
> **Comments on the computational burden**
>
> Thank you for your time and comments. We address the computational cost and answer your questions in the following:
>
> Computational cost
> The paper has been updated, and we have added a small paragraph in the section “Limitations,” addressing the computational burden of GPs and FBGP in general.
>
> It is well-known that Gaussian Processes do not scale well, and it is an even bigger problem for fully Bayesian GPs (FBGPs). However, while we acknowledge its existence, we do not see the computational cost as a significant drawback. The reason is two-fold:
>
> 1. First, we study a problem that mainly appears when the data set is quite small. Very often, in active learning applications with GPs, the datasets are smaller than 1000 data points; In such scenarios, the proposed approach should work. Nevertheless, even with 1000 data points, MCMC inference can be very heavy to compute. In this case, we are convinced that this computational cost is relatively small - or even insignificant - when compared to the cost of querying a new data point (with real simulators, the cost of querying a point is expensive and is often minutes and sometimes even be hours [1,2,3]). Hence, there is clearly a tradeoff between the cost of FBGP and the cost of acquiring new points from the underlying simulators. Notice that the more points we decide to acquire, the more simulation runs we need to perform. FBGP aims at maximizing information acquisition while minimizing the simulation costs at the same time. For this reason, we believe that the tradeoff mentioned above generally favours our approach, particularly when applied within computationally heavy simulation setups.
>
> 2. On the other hand, let us say that the proposed method does disproportionally use a significant amount of time. Then, other methods that can be employed to reduce the associated computational cost:
>
>     i. Since the proposed acquisition functions utilize the disagreement among modes of the hyperparameters, one could switch to a regular GP when the disagreement is no longer present. This is usually the case for the first few iterations. When the active learning process stabilises, the added value of FBGP is reduced.
>
>     ii. One can use Batch Active Learning to evaluate the FBGP less times. Instead of single-point queries, batch of points can be used to cut down on the evaluation time. If we assume, for example, a batch number of 10 points, the general evaluation cost would be reduced reduced significantly. Still, the batch size also has an important impact on the tradeoff mentioned before, since larger batches will inevitably lead to higher simulation costs per iteration or query. In order to somehow mitigate this cost, or, at least, to take advantage of it, it is important to take batch heterogeneity (non-redundancy, high information, etc.) into account, and this is another research field on its own.

---

> ### Author Response · Authors · 2022-08-02
> **Answer to questions**
>
> 1. Yes, the most reasonable mode is generally given by the mode where the marginal likelihood is maximized. That is also why we used it when we made the predictions. However, the marginal likelihood is maximized to the current data, and as we add more data, that specific mode might move/jump around. We can say that the mode is 'biased' toward the current training set. With very few data points, it can hinder the performance of active learning if only the most reasonable mode is used.  Take, for example the Gramacy simulator in Figure 3. If only a few data points are present, the most reasonable mode will model the data as a non-wiggly function. However, as we show in Figure 4, the performance of ALM (only considering the most reasonable mode) and QB-MPG (considering multiple modes) is very different. Thus, when the data changes in every iteration, the best mode is not necessarily the mode that maximizes the marginal likelihood in the first iteration.
>
> 2. We have slightly rewritten the explanation and added a figure in the appendix, showing the spreading of the data points queried by the two acquisition functions. In the first iterations (0-20), B-QBC queries many points in the middle due to the variance in the mean function that arises because of the two competing modes, cf. Figure 6. Oppositely, the B-ALM component encourages exploration since it aims to reduce the overall predictive uncertainty. This is seen in the spreading of the queried data points that already cover a good portion of the space after 10 iterations. After 40 iterations, both acquisition functions have explored the space similarly.
>
> 3. Yes, you are right. \mu_GMM is equivalent to \bar{\mu}, and thus the only difference between B-QBC and QB-MGP is the expected variance of the Mixture of Gaussian Processes. In most cases, they have similar performance, but in some cases, there are indeed quite different. Generally, B-QBC reduces the marginal likelihood, whereas QB-MGP is better in reducing the RMSE. We believe that since B-QBC is mainly focusing on disagreement the modes, it is more related to the marginal likelihood, whereas the RMSE is often reduced best by spreading in the data points, which is precisely what the variance encourages.
>
> [1] Chabanet, Sylvain, Hind Bril El-Haouzi, and Philippe Thomas. "Coupling digital simulation and machine learning metamodel through an active learning approach in Industry 4.0 context." Computers in Industry 133 (2021): 103529.
>
> [2] C. Riis, F. Antunes, G. Gurtner, F. C. Pereira, L. Delgado, and C. M. Lima Azevedo. Active learning metamodels for atm simulation modeling. In Proceedings of the 11th SESAR Innovation Days, 2021
>
> [3] Gorissen, Dirk, et al. "Automatic approximation of expensive functions with active learning." Foundations of Computational, Intelligence Volume 1. Springer, Berlin, Heidelberg, 2009. 35-62.

---

> > ### Comment · Reviewer_8dtu · 2022-08-08
> > **Response to Rebuttal**
> >
> > Thank you for the detailed responses - I am happy with the answers for the most part. In my opinion, it is important to incorporate in the final paper 1) the discussion around the compute demands of FBGP in the final paper and 2) highlight that B-QBC and the MGP method yield the same overall model means, so \mu_GMM is the same as \bar{\mu} and it is the expected variance which is different hence the performance difference has to be explained accounting for that.
> >
> > I am happy to raise my contribution score to 3 and overall score to 7.

---

### Official Review · Reviewer_rWt1 · 2022-07-12

**Rating:** 5
**Confidence:** 4
**Soundness:** 3 good
**Presentation:** 3 good
**Contribution:** 3 good

**Summary:**

This paper focuses on active learning using Gaussian Processes, where the authors take a fully Bayesian approach to optimize the bias-variance trade-off, an important problem in various learning problems.
Especially, this work focuses on a small-data setting, where such bias-variance trade-off can significantly affect the learning outcomes.
The authors propose two acquisition functions based on FBGP for regression tasks, showing the potential advantages of the proposed schemes.


**Questions:**

1. Please elaborate on the major advantages of the proposed active learning schemes in a small-data setting, especially, based on the results shown in Figure 4, since it is unclear based on the current evaluation results.

2. Please compare the computational cost of the proposed methods, in comparison with other schemes considered in this study.

3. Please clarify whether and how the proposed schemes can be generalized beyond the tasks/settings considered in the current study.

4. How would the proposed schemes (especially, QB-MGP) compare to other active learning schemes (such as ELR) that focus on predictive performance?

5. Please clarify what would be the main potential advantages of the proposed schemes compared to other recent developments in Bayesian active learning.






**Limitations:**

The authors note that they "create a generic active learning acquisition function" and therefore "there is no direct negative societal impacts."


**Strengths And Weaknesses:**

Taking a Bayesian approach to automatically balance the bias-variance trade-off and improve the robustness of the learning outcomes and the predictive performance of the learned models in a small-data setting is expected to be beneficial, and this work demonstrates the potential advantages of using FBGP  for active learning in regression tasks.

1. Table 2 and Figure 4 show that the two proposed acquisition functions - B-QBC and QB-MGP - can lead to better regression results for a number of test cases. However, these results also show that the proposed scheme do not consistently outperform other alternatives, nor significantly improve the learning outcomes when they do. Especially, while the comparison based on AUC (in Table 2) make the performance improvement attained by the proposed schemes relatively prominent in some cases, the actual improvement shown in the curves (Figure 4) do not appear to be very significant. This is especially so when the data size is relatively small, even though that is the setting that motivates the current work.

2. The computational burden of taking a fully Bayesian approach is not discussed in detail, and the authors simply mention that they assume that the computational burden for querying new data points would far exceed that for the Bayesian inference. While this may be the case in some applications, it would be nevertheless important and informative to compare the computational cost of different active learning schemes, which is absent in the current study.

3. The authors do not discuss the general applicability of the proposed scheme beyond the regression tasks considered in the current work. For example, BALD - shown to outperform the proposed scheme in a number of cases - is widely used for active learning for classification, and it would be meaningful to present how the proposed schemes would be applied to classification tasks and how their performance would compare to BALD and other alternatives.

4. While the authors proposed QB-MGP to consider the "predictive performance" of the learned model, when the prediction performance is of main interest, it would be more sensible to compare the performance with other active learning schemes that focus on this aspect. For example, ELR (expected loss reduction) strategies are widely used to acquire new data points that are expected to optimally reduce the error rather than reducing the uncertainty of the model parameters.

5. Recently, a number of Bayesian active learning schemes have been proposed based on the ELR strategy, whose performance has been shown to outperform BALD and various other methods with theoretical convergence guarantees. Some recent examples include:

(1) Tan et al, Diversity Enhanced Active Learning with Strictly Proper Scoring Rules, NeurIPS 2021.
(2) Zhao et al, Efficient Active Learning for Gaussian Process Classification by Error Reduction, NeurIPS 2021.

Considering that the aforementioned Bayesian active learning methods have shown to consistently outperform BALD and other existing methods with the added benefit of convergence guarantee to the optimal model, the potential benefits of the proposed schemes remain somewhat unclear. Further comparison and elaboration would be necessary to clarify the most significant advantages of the proposed methods, in comparison with the latest relevant developments in Bayesian active learning.

---

> ### Author Response · Authors · 2022-08-02
> **Answers to questions**
>
> Thank you for your time and comments. We answer your questions in the following:
>
> 1. We do understand your point. The main takeaway from Table 2 and Figure 4 is that there is no optimal active learning that is able to consistently outperform the others, independently of the simulator. One of the reasons is that the performance of the active learning strategy is strongly associated with the characteristics of the simulation output space (regularity, homoscedasticity vs heteroscedasticity, periodicity, etc.). In this work, we tried to choose not only benchmark functions that are commonly used in the related literature, but also those with different characteristics that could be shared by real expensive-to-run simulators originated from different fields. It can be quite challenging to predict, a priori, how the output of an arbitrary simulator will look like. Therefore, we argue that, all things considered, the proposed approach is a safe option when minimal or no knowledge about the simulator is available, which is often the case. Thus, using the proposed acquisition functions will be a safer choice.
>
> 2. We have added a small paragraph in the section “Limitations,” addressing the computational burden of GPs and FBGP in general. There are no significant differences in the computational cost between the acquisitions used per se, only in the underlying model. ALM can be used with a GP, where the others need the MCMC samples. The computational cost of that is well studied in the literature.
>
> 3. We agree that active learning for regression and classification are related but they constitute different topics of research. Our work specifically focus on regression settings, which according to [1], falls behind (w.r.t. classification models) in terms of the available research and number of contributions. The proposal of FBGP goes in line with addressing this gap, as it is especially tailored for regression tasks.
>
> 4. We do compare our proposed methods with methods that focus on the predictive performance. Both ALM and B-ALM aim at reducing the predictive uncertainty, which can be seen as a proxy for reducing the expected loss. Please refer to [2] and [3] on this issue.
>
> 5. Thank you for the references. Both papers present some interesting active learning acquisition functions for classification tasks, and it would be great if they could be applied in a regression setting. Although they both mention “expected loss of reduction,” we do not see how we can use it in a regression setting.
>
> [1] Kumar P, Gupta A. Active learning query strategies for classification, regression, and clustering: A survey. JOURNAL OF COMPUTER SCIENCE AND TECHNOLOGY 35(4): 913–945 July 2020. DOI 10.1007/s11390-020-9487-4
>
> [2] Burr Settles. Active learning literature survey. Computer Science Technical Report 1648, University of Wisconsin-Madison. 2009.
>
> [3] Zhao et al, Efficient Active Learning for Gaussian Process Classification by Error Reduction, NeurIPS 2021

---

> > ### Comment · Reviewer_rWt1 · 2022-08-09
> > **Response**
> >
> > Thank you for the response to the review comments.
> >
> > 1. The rebuttal to the first question is reasonable, and I understand the authors' point. It would be great if the authors could include the relevant discussion in the manuscript, either in the discussion/conclusion sections or as a potential limitation.
> >
> > 2. Thank you for the response and the including a brief discussion regarding the computationa cost.
> >
> > 3. I agree that technically, active learning for regression and classification respectively involves related but different challenges, despite the high-level similarity in terms of the objective.
> > Please consider incorporating your response to the question into the main text.
> >
> > 4. Thanks for the clarification regarding the emphasis on predictive performance.
> > At least a brief remark on their relation to ELR would be beneficial for readers.
> >
> > 5. Thank you for the response. I understand the authors' point - which is also related to question 3 and your response to the previous question.
> >
> > Overall, I am happy with the authors' response and have adjusted the score accordingly.
> > Thank you.

---

### Official Review · Reviewer_u8Av · 2022-07-12

**Rating:** 6
**Confidence:** 2
**Soundness:** 3 good
**Presentation:** 3 good
**Contribution:** 2 fair

**Summary:**

The authors propose a novel active learning scheme for GPs based on a full-Bayesian solution.

**Questions:**

I have only a suggestion: to complete the  state-of-the-art discussion,
 including the acquisition functions in GP schemes (for regression of for quadrature) that considers also the gradient information as suggested in

D. H. Svendsen, et al. Active Emulation of Computer Codes with Gaussian Processes - Application to Remote Sensing, Pattern Recognition Volume 100, 2020,

F. Llorente, et al, "Adaptive quadrature schemes for Bayesian inference via active learning", IEEE Access, Volume 8, 2020.

M. Kanagawa and P. Hennig, “Convergence Guarantees for Adaptive Bayesian Quadrature Methods,” in Advances in Neural Information Processing Systems, 2019, pp. 6234–6245.

**Limitations:**

 The contribution seems a bit incremental.

**Strengths And Weaknesses:**

The paper is well-written and it is nice to read this work. This is the the main strength, in my opinion.
The state-of-the-art discussion is the nicest part, in my opinion.

Weakness: the contribution seems a bit incremental.

---

> ### Author Response · Authors · 2022-08-02
> **We have completed the state-of-the-art discussion**
>
> Thank you for your time and comments, and thank you for referring to relevant literature.
>
> The paper has been updated, and we have completed the state-of-the-art discussion by including the two papers on model-based acquisition functions using the function values and gradients in the active learning acquisition functions [Fernandez et al. 2020, Svendsen et al. 2020].
>
> We did not include the work from M. Kanagaw and P. Henning [2019] since we think that their methods and acquisition functions are partly covered by the work of Fernandez et al. [2020] and Svendsen et al. [2020] and is more specialised for quadrature methods.

---

### Meta-Review · Area_Chair_6UBr · 2022-08-28

**Recommendation:** Accept
**Confidence:** Certain

**Metareview:**

This paper took a “fully Bayesian” perspective on Gaussian process (FBGP) regression tasks. The key technical contribution builds on the argument that the optimal mode of the hyperparameter posterior (i.e. over the length-scale and the noise parameters) corresponds to the optimal bias-variance trade-off. One would expect that a fully Bayesian approach (with a reasonable prior) would indeed be beneficial; it is interesting and useful to understand how such methods perform in practice in terms of robustness and the (extra) computational burden. All reviewers agree that this work provided sufficient empirical support that demonstrates the potential advantages of using FBGP for active learning in regression tasks. During the rebuttal/discussion phase, the authors provided extra empirical evidence which makes the results more convincing (e.g., discussion and empirical analysis on the computational complexity and the factors that affect such). There were no other critical concerns in the reviews.

There are valuable suggestions in the reviews, including improving the clarity when analyzing the results against baselines, providing details of the experimental setting and results, and an in-depth discussion of the computational cost (which appears to be a key message to convey) The authors are strongly encouraged to address the concerns raised in the reviews when preparing a revision of this paper.


**Award:**

No

---

### Decision · Program_Chairs · 2022-09-14

Accept